Field-measured drag area is a key correlate of level cycling time trial performance

Peterman James E.
Lim Allen C.
Ignatz Ryan I.
Edwards Andrew G.
Byrnes William C. Byrnes@Colorado.edu
Department of Integrative Physiology, University of Colorado Boulder , Boulder, CO , USA
Zhao Wenbing
Electronic publication date: 2015 Aug 11
Publication date: 2015
Volume: 3
Electronic Location ID: e1144
Received 2015 May 11; Accepted 2015 Jul 10
Copyright: © 2015 Peterman et al.
Copyright year: 2015
Copyright holder: Peterman et al.
License: This is an open access article distributed under the terms of the Creative Commons Attribution License, which permits unrestricted use, distribution, reproduction and adaptation in any medium and for any purpose provided that it is properly attributed. For attribution, the original author(s), title, publication source (PeerJ) and either DOI or URL of the article must be cited.
License URL: https://creativecommons.org/licenses/by/4.0/

Keywords: Drag coefficient, Power meter, Field testing, Power output, Predicting performance, Exercise physiology

Funding: CycleOps PowerTap This research was funded by CycleOps PowerTap. The views expressed are those of the authors and do not reflect those of CycleOps PowerTap. The funders had no role in study design, data collection and analysis, decision to publish, or preparation of the manuscript.

==============================
Drag area (Ad) is a primary factor determining aerodynamic resistance during level cycling and is therefore a key determinant of level time trial performance. However, Ad has traditionally been difficult to measure. Our purpose was to determine the value of adding field-measured Ad as a correlate of level cycling time trial performance. In the field, 19 male cyclists performed a level (22.1 km) time trial. Separately, field-determined Ad and rolling resistance were calculated for subjects along with projected frontal area assessed directly (AP) and indirectly (Est AP). Also, a graded exercise test was performed to determine V˙O2 peak, lactate threshold (LT), and economy. V˙O2 peak (lmin−1) and power at LT were significantly correlated to power measured during the time trial (r = 0.83 and 0.69, respectively) but were not significantly correlated to performance time (r = − 0.42 and −0.45). The correlation with performance time improved significantly (p < 0.05) when these variables were normalized to Ad. Of note, Ad alone was better correlated to performance time (r = 0.85, p < 0.001) than any combination of non-normalized physiological measure. The best correlate with performance time was field-measured power output during the time trial normalized to Ad (r = − 0.92). AP only accounted for 54% of the variability in Ad. Accordingly, the correlation to performance time was significantly lower using power normalized to AP (r = − 0.75) or Est AP (r = − 0.71). In conclusion, unless normalized to Ad, level time trial performance in the field was not highly correlated to common laboratory measures. Furthermore, our field-measured Ad is easy to determine and was the single best predictor of level time trial performance.

Introduction

A cyclist’s ability to produce and sustain mechanical power output is highly dependent upon physiological characteristics, particularly V˙O2 max, lactate threshold (LT), and economy (Coyle et al., 1988; Coyle, 1995). In the laboratory, where resistive forces are controlled or minimized, these physiological factors have been successfully used to predict simulated time trial performance (Coyle et al., 1988; Loftin & Warren, 1994; Coyle, 1995; Bishop et al., 2000; Bentley et al., 2001; Lamberts et al., 2012). However, in the field when cycling on level terrain at constant velocities (>40 km hr−1), more than 90% of the total resistance (RTOT) impeding the forward motion of a bicycle-rider system is determined by aerodynamic resistance (Ra) (Pugh, 1974; Di Prampero et al., 1979; Martin et al., 1998; Olds & Olive, 1999). Thus, measures of a cyclist’s ability to supply mechanical power do not always predict performance time in time trial racing (Hoogeveen & Schep, 1997; Balmer, Davison & Bird, 2000). For example, Balmer, Davison & Bird (2000) demonstrated that while peak mechanical power output assessed during a graded exercise stress test does correlate highly (r = 0.99, p < 0.001) with average mechanical power output during a 16.1 km field time trial, neither the laboratory peak mechanical power output nor the average mechanical power output during the time trial correlated well with performance time (r = 0.46, p > 0.05). One possible explanation for these results is that the resistance impeding the forward motion faced by competitive cyclists is variable enough that mechanical power output alone may not predict performance (Jeukendrup & Martin, 2001).

Measuring the aerodynamic resistance force (Ra) during cycling can be complex. Direct measures include wind-tunnel tests (Davies, 1980; Kyle, 1991; Martin et al., 1998), motorized towing (Di Prampero et al., 1979; Capelli et al., 1993), and coasting deceleration methods (De Groot, Sargeant & Geysel, 1995; Candau et al., 1999). Even though these direct measurements are accurate, they are impractical for most researchers and practitioners. For example, wind-tunnel testing facilities are inaccessible and costly. In addition, during wind tunnel testing, the bikes are not physically moving through the air and the pedaling motion introduces noise in the force measurement system during each pedal stroke. Motorized towing and coasting deceleration methods are also elaborate in their setup and execution meaning they are not an ideal substitute to a wind tunnel.

Because measuring the aerodynamic resistance force can be complex, it is sometimes assumed to be directly proportional to measures or estimates of the projected frontal area (AP) of the bicycle and rider (Di Prampero et al., 1979; De Groot, Sargeant & Geysel, 1995; Olds & Olive, 1999; Heil, 2001; Anton et al., 2007). This assumes, though, that between individuals, aerodynamic resistance changes predictably with changes in AP. However, evidence exists to the contrary. Previous research shows a lack of proportionality between an individual’s measured AP and aerodynamic resistance (Kyle, 1991; De Groot, Sargeant & Geysel, 1995). This discrepancy must be due to variability in the coefficient of drag (Cd, see equation (2)), which is influenced by the shape of the bicycle and rider, varies greatly between individuals, and does not change proportionally with changes in projected frontal area (Debraux et al., 2011).

Recently, it has been demonstrated that measuring mechanical power output and speed in the field with cycle mounted power meters is a viable and accessible technique for determining an individual’s aerodynamic and rolling resistances (Martin et al., 2006; Debraux et al., 2011; Lim et al., 2011). This technique, in combination with standard physiological profiling, could improve our ability to predict field performance since it has been argued that physical factors resisting forward motion play a larger role in performance outcome than physiological variables (Jeukendrup & Martin, 2001). Anton et al. (2007) have previously shown that projected frontal area alone is correlated to performance time during a level time trial (r = − 0.73). However, their estimated frontal area did not improve the prediction of level time trial performance compared to just maximal mechanical power output measured in the lab. As described above, AP is not directly proportional to aerodynamic resistance. Thus, the aerodynamic characteristic of a cyclist (drag area (Ad)), which includes both AP and Cd, may provide a better correlation for time trial performance.

Accordingly, the purpose of this study was to quantify physiological determinants of endurance performance in conjunction with physical factors that contribute to resistance during cycling. Correlations between the various determinants/factors and level time trial performance were then compared. We hypothesized a cyclist’s physiological capacity or average power output would not predict level time trial performance time unless normalized to some representation of aerodynamic resistance. Additionally, we hypothesized that estimates of aerodynamic resistance, such as direct and indirect assessments of projected frontal area, would not correlate to performance as highly as the actual aerodynamic characteristic of the cyclist.

Methods

Subjects

Nineteen competitive male cyclists volunteered for this study. All subjects were licensed cyclists (United States Cycling Federation Category Pro/1/2 Road or Pro/Expert MTB) living, training, and racing in the Colorado, USA area for a minimum of two months. All tests were performed during the height of the local competitive cycling season. Descriptive characteristics of our nineteen subjects and the results from the laboratory testing are presented in Table 1. All subjects were informed of the risks involved with participation in the study and gave written informed consent before participating. The Human Research Committee at the University of Colorado Boulder approved the protocol used for this study (reference number 0600.20).

Table 1 Mean ± SD, minimum, and maximum descriptive information for all 19 subjects and values for the primary physiological performance measures assessed during the laboratory graded exercise stress test.

	Mean ± SD	Min	Max	
Age (years)	27.6 ± 4.6	20	36	
Height (cm)	174.0 ± 6.0	164	184	
Body mass (kg)	70.0 ± 8.0	59.5	87.3	
Body fat (%)	10.4 ± 2.64	6.0	16.5	
Bone density (g cm−2)	1.22 ± 0.11	1.03	1.41	
Years racing	8 ± 5	3	20	
VO2 peak (l min−1)	4.67 ± 0.40	3.91	5.47	
VO2 peak (ml kg−1 min−1)	67.6 ± 6.4	54.2	76.5	
Power at VO2 peak (Watts)	362 ± 30	308	410	
Power to Mass at VO2 peak (W kg−1)	5.15 ± 0.51	4.21	5.95	
LT 1 mM as % of VO2 peak	76 ± 4.5	65	82	
Power at LT 1 mM (Watts)	271 ± 29	230	331	
Power to Mass at LT 1 mM (W kg−1)	3.87 ± 0.48	3.03	4.53	
Economy Wl O2−1	73.5 ± 3.1	69.1	78.9	
Max heart rate (bpm)	183 ± 8	168	197	

Power measuring devices and calibration

All subjects utilized a rear hub power meter (CycleOps PowerTap, Madison, Wisconsin, USA) set to record external power output, ground velocity, cadence, and time measured at a frequency of 61 Hz with data averaged and recorded in epochs of 1.26 s for all variables. Heart rate was recorded using a CycleOps heart rate monitor chest strap that transmitted to the power meter computer. Nineteen distinct power meters were used throughout the study, one for each subject. Each subject used the same power meter for the time trial, laboratory testing, and aerodynamic/rolling resistance measures. Prior to commencing trials, all units used during testing were calibrated against a zero torque reference while pedals were stationary and unloaded as directed by the manufacturer.

Laboratory measures and protocol

Laboratory measured endurance performance predictors including peak oxygen consumption (V˙O2 peak), lactate threshold (LT), and economy (WattslO2−1min−1 at LT) were measured during a graded exercise test conducted within three days of the subject’s time trial. Each test was performed on the subject’s personal road bicycle attached to an electronically braked trainer (CompuTrainer®, Seattle, Washington, USA) with power measured using the rear hub power meter loaned to the subject at the beginning of the study. The protocol began at a power output between 100 and 150 watts and increased by approximately 30 watts every 4 min until volitional fatigue. All tests were conducted at an altitude of 1,625 m (5,330 ft) with an average barometric pressure, temperature, and humidity of 630.5 ± 3.4 mm Hg, 22.9 ± 1.4 °C, and 37.1 ± 6.9%, respectively. Additionally, a fan was used to cool subjects throughout each exercise stress test.

Oxygen consumption and carbon dioxide production were averaged every 15 s through computer assisted indirect calorimetry using Parvomedics software and hardware to integrate input from a Validyne pressure tranducer and Perkin Elmer mass spectrometer. The pressure transducer was linked to a Hans Rudolph pneumotach measuring inspired ventilation and the Perkin Elmer mass spectrometer sampled from a 4-liter mixing chamber. Calibration procedures are summarized in Appendix S1.

Peak oxygen consumption was defined as the highest rate of oxygen consumption for a sampling interval of one minute during the graded exercise stress test. Subjects were vigorously encouraged to give a maximal effort and in all cases exceeded a respiratory exchange ratio of 1.15 and a blood lactate of 7 mM at volitional exhaustion. For the measure of economy, only the oxygen consumed over the last two minutes of each 4-minute stage was used to ensure steady state measures. Economy was measured as the ratio between power output and oxygen consumption WattslO2−1 at the lactate threshold from a regression of the oxygen consumption versus power relationship. In all subjects, the relationship between oxygen consumption and power output was linear (r > 0.99) through the penultimate stage.

Blood lactate concentration was measured at rest and over the last minute of each 4-minute stage. For each sample, approximately 50 µl of blood was drawn via finger pricks into a 75 µl capillary tube. Twenty-five µl was then mixed with a “cocktail” containing 50 µl of a buffer, lysing (Triton XL-100), and anti-glycolytic (sodium fluoride) solution. Lactate was finally analyzed using a YSI 2300 lactate analyzer (Yellow Springs, Ohio, USA). Before each test, the lactate analyzer was calibrated against a known standard, and re-calibrated every 15 min. The lactate threshold was defined as a point 1 mM above a baseline that included resting lactate (Coyle et al., 1988). This process includes some subjective interpretation in determining the baseline values leaving room for potential human error. Thus, the average value for these points determined from eight independent observers was used in data analysis. No significant difference was found between observers.

Heart Rate during laboratory testing was measured using radio telemetry (Polar®, Lake Success, New York, USA) each minute while perceived exertion was measured using the Borg 6–20 scale 3 min into each stage. Within two days of the laboratory test, body composition was assessed using dual energy X-ray absorbtiometry (GE LUNAR DXA system, Fairfield, Connecticut, USA).

Time trial

The level time trial was conducted over two laps of a four-corner loop in Hygiene, Colorado. The total distance of the time trial was 22.1 km. Over the course of each lap, subjects gained and lost 79 m of elevation for a net elevation gain of 0 m. The layout of the course allowed the subjects to ride continuously with no stop signs or traffic lights impeding their effort. Subjects were allowed to utilize time trial bicycles equipped with time trial bars and an aerodynamic front wheel. Six subjects rode their standard road bike with no additional aerodynamic equipment. Another four subjects rode their standard road bike equipped with aerodynamic handlebars. The other nine subjects utilized a time trial bicycle equipped with aerodynamic handlebars with an aerodynamic deep dish or three-spoked front wheel.

Within two weeks prior to the time trial, subjects were asked to pre-ride the course to familiarize them with the route. All subjects had previous experience training on the selected course. Directly prior to the time trial, subjects performed their personal pre-competition warm-up. At this time, tires were inflated to 8.3 bar with the riders off the bicycle. Immediately before the time trial, the subject’s individual mass (MR) and bicycle rider system mass (MBRS) were measured using an electronic scale previously calibrated against a laboratory balance scale (Detecto Scales, Webb City, Missouri, USA). At the start line, the on-board computers were cleared and the power meters calibrated against a zero load. During the time trial, subjects were blinded from viewing their power output but allowed to view speed, time, distance, cadence, and heart rate. In addition to the time measured by the on-board computer, performance time was also measured using an external stopwatch.

All time trials were held between 9 and 11 am. Air density was calculated from measures of ambient temperature, station pressure, and relative humidity collected with a Vantage Pro model weather instrument (Digital Instruments, Enterprise, Oregon, USA). Though an attempt was made to schedule as many subjects as possible on the same day to control for wind, environmental conditions, and the competitive atmosphere, a total of ten separate time trials were performed with seven subjects performing the time trials alone and the others on one of three occasions. On days in which multiple subjects performed the time trial, enough time was given between the start of each subject so that subjects were not able to pace or draft off each other during the trial. During these separate occasions, the time trials were allowed to proceed as long as the wind did not exceed a 3 or a “gentle breeze” on the Beaufort Wind Scale. This is equivalent to a wind speed less than 19 km hr−1 characterized by surroundings in which smoke rises vertically (0) to a wind velocity were leaves and small twigs constantly move (3).

Data reduction and analysis

Immediately after the time trial, data collected from the CycleOps power meter and heart rate monitor was downloaded from the onboard computer to an Apple G4 computer. Downloaded data included time, power output, speed, cadence, and heart rate in 1.26-second intervals. Using Power Coach™ (Kochli Sport, Sonvilier, Switzerland) software operated by Apple G4 computers, the performance time measured by the external stopwatch was located on the data and isolated. The distance from this isolated data was then checked to ensure that it matched the actual distance of the course. The data isolated in this manner did not vary by more than 50 m (7.26 ± 4.26 s) from the actual distance of the course. Each line of data was also checked for any potential recording problems with any erroneous (loss of signal or supra-physiological) data interpolated between the adjacent data points. On average, approximately four seconds of data were interpolated in any given time trial (range = 0–24 s), with never more than 2 erroneous data points occurring in sequence. Finally, the statistics of interest were calculated using specific code written for Matlab® (Mathworks Inc., Berkeley, CA, USA).

Field measured aerodynamic and rolling resistance profiling

Within three weeks of the time trial, aerodynamic and rolling resistances were determined from field measures of power and velocity (Edwards & Byrnes, 2007; Lim et al., 2011). Using the methodology described by Di Prampero et al. (1979), drag area (Ad) and rolling resistance (Rr) were calculated for each subject (see Appendix S1).

Projected frontal area

Projected frontal area was determined based on the recommendations of Olds & Olive (1999). Digital photographs (Nikon Cool Pix 4300, Melville, New York, USA) of the subjects were taken while subjects sat in a riding position on the bicycle they used during the time trial, which was mounted on a stationary trainer. Photographs were analyzed using NIH Image 1.62 software. This software, after calibration against a known distance or area, automatically calculates the area of a given tracing. Projected frontal area was calculated from tracings of the subject’s body and helmet only with the bicycle excluded. In addition to actual measures of projected frontal area, estimates of projected frontal area were also made. Numerous equations have been provided in the literature to either predict or account for some aspect of the aerodynamic drag or frontal area of cyclists (Pugh, 1974; Davies, 1980; McLean, 1993; Olds, Norton & Craig, 1993; Olds et al., 1995; Heil, 2001; Heil, 2002). Some of the equations rely on the assumption that AP is a constant fraction of body surface area (BSA), while others derive more complex algorithms to predict AP based on various anthropometric qualities of a cyclist (i.e., height and weight). A total of ten different equations from previous research (DuBois & DuBois, 1916; Pugh, 1974; Davies, 1980; McLean, 1993; Olds, Norton & Craig, 1993; Olds et al., 1995; Heil, 2001) were compared.

Statistical analyses

Bivariate correlations were performed to assess the relationship between physical, physiological, and normalized variables to time and average power measured during the time trial. In addition, stepwise multiple regressions were performed to locate the best single or combination of variables predictive of time and average power output. Differences between correlation coefficients were then located using the Hotelling test. The Pearson product-moment correlation coefficients were used to determine the relationship between actual and predicted values of Ap and Ad. Repeated-measure ANOVAs were run to determine if mean differences existed between actual and predicted values of Ap. If significance was found, the more conservative Scheffe post-hoc test was used to determine which variables differed. Significance for all calculations was set at p < 0.05. Descriptive data are reported as the mean, standard deviation, minimum, and maximum.

Results

The primary components of resistance during cycling are presented in Table 2. Because the AP was calculated with the rider only while the Ad was calculated for the entire bicycle and rider system, it would be technically incorrect to calculate a Cd for the bicycle and rider system or rider alone. Still, dividing Ad by the AP would give a estimated Cd of 1.03 ± 0.12. A correlation of 0.735 was found between AP and Ad. No relationship, however, was found between AP and the Cd (r = − .170, p = 0.486).

Table 2 Primary determinants of aerodynamic and rolling resistance measured on the bicycle and body position used during the time trial.

	Mean ± SD	Min	Max	
Air Density (kg m−3)	1.00 ± 0.02	0.98	1.05	
k constant (N m−2 s2)	0.170 ± 0.028	0.126	0.228	
Ad (m2)	0.349 ± 0.059	0.258	0.462	
AP (m2)	0.338 ± 0.050	0.272	0.444	
Est AP (m2) (Olds et al., 1995)	0.342 ± 0.017	0.32	0.375	
Est AP (m2) (Heil, 2002)	0.327 ± 0.027	0.293	0.386	
Rr (N)	4.88 ± 1.27	2.71	7.39	
Cr (dimensionless)	0.006 ± 0.002	0.004	0.009	
Total Mass (kg)	80.33 ± 9.42	68.9	99.7	
Notes.

Constant k aerodynamic character of a cyclist

Ad drag area

AP measured projected frontal area

Est AP estimated projected frontal area

Rr rolling resistance

Cr quality of the tire and road interface

total mass mass of bicycle and rider

The results of the time trial are presented in Table 3. Of the physiological variables measured, V˙O2 peak (l min−1) (r = 0.83, p < 0.001), power at V˙O2 peak (r = 0.67, p < 0.001), and power at LT (r = 0.69, p < 0.001) were significantly correlated to field-measured power output during the time trial. These physiological measures, however, were not strongly or significantly related to time trial performance time (V˙O2 peak r = − 0.42, p = 0.08; power at V˙O2 peak r = − 0.43, p = 0.07; power at LT r = − 0.45, p = 0.06). In addition, although significant, the field-measured power output during the time trial was correlated to performance time with an r-value of only −0.59. Thus, as hypothesized, non-normalized physiological measures were not strongly related to time trial performance time.

Table 3 Performance and environmental variables measured during the time trial.

	Mean ± SD	Min	Max	
Total time (min:sec)	31:24 ± 2:15	28:05:00	34:52:00	
Temperature (°C)	26.2 ± 4.5	20	35	
Humidity (%)	30.7 ± 9.5	14	50	
Air pressure (mmHg)	629.2 ± 3.7	626.4	638.3	
Air density (kg m−3)	0.97 ± 0.02	0.95	1	
Average power (Watts)	303 ± 26	259	354	
Standard deviation of the power during the time trial (Watts)	80 ± 14	61	113	
Power (Watts kg−1)	4.32 ± 0.44	3.27	4.98	
Power (% of LT)	112.4 ± 9.2	96.1	133	
Power (% of VO2 peak)	84.1 ± 5.9	69.6	92.5	
Heart rate (bpm)	173 ± 6	161	183	
Standard deviation of the heart rate during the time trial (bpm)	8 ± 2	4	11	
Heart rate (% of LT HR)	111.9 ± 3.8	101.5	117.6	
Heart rate (% of Max HRr)	95.5 ± 1.1	93.3	96.8	
Max heart rate during the time trial (bpm)	180 ± 6.3	167	192	

The correlation of physiological measures and performance time greatly increased when normalized to aerodynamics. When normalized to Ad, power at V˙O2 peak (watts m−2) (r = − 0.92, p < 0.001) and power at LT (r = − 0.85, p < 0.001) were better correlated to performance time (Fig. 1). Other normalized laboratory physiological performance measurements were either not significant or if significant were only moderately correlated. The correlation to performance time was best when mean field-measured power output was normalized to either k (r = − 0.92, p < 0.001) or Ad (r = − 0.92, p < 0.001). Figure 2 displays the relationship between performance time relative to the mean field-measured power output as well as mean field-measured power output normalized to body mass, AP, and Ad. Rolling resistance alone or when used to normalize power or physiological measures was not related to performance time.

Figure 1 The relationship between time trial performance time relative to non-normalized and normalized laboratory measures.

(A) Performance time relative to power at V˙O2 peak. (B) Performance time relative to power at lactate threshold (LT). (C) Performance time relative to power at V˙O2 peak normalized to field-determined drag area (Ad). (D) Performance time relative to power at LT normalized to field-determined Ad.

Figure 2 The relationship between time trial performance time relative to the non-normalized and normalized field-measured power output.

(A) Performance time relative to mean field-measured power output. (B) Performance time relative to mean field-measured power output normalized to body mass. (C) Performance time relative to mean field-measured power output normalized to frontal area (AP). (D) Performance time relative to mean field-measured power output normalized to field-determined drag area (Ad).

Both k (r = − 0.32, p = 0.18) and Ad (r = − 0.32, p = 0.17) were not correlated to field-measured power output during the time trial. However, both k (r = 0.85, p < 0.001) and Ad (r = 0.85, p < 0.001) alone were significantly related to performance time. Although significant (p = 0.04), measured AP alone had a lower correlation (r = 0.47) and no methods for determining Est AP correlated with performance time (r = 0.010 to 0.099). Furthermore, field-measured power output normalized to AP was only modestly correlated to performance time (r = − 0.75).

Assuming that all of the resistance faced by the cyclists during the time trial was due to aerodynamic and rolling resistance, estimates of power for each component are given. Aerodynamic power was calculated from the k measured for each subject’s bicycle position while rolling resistance was assumed to be the remainder of the power. During the time trial, aerodynamic and rolling resistance accounted for 89.77% and 10.23% of the total power at 272 ± 23 and 31 ± 12 watts, respectively.

Except for the Olds et al. (1995) and Heil (2001) methods (P > 0.999 and 0.322, respectively), the mean value for AP (0.338 ± 0.049 m2) was significantly over predicted by all other estimations of AP as a percent of BSA (p < 0.001; Pugh (34), P = 0.011). The range for Est AP as a percent of BSA was 0.306 ± 0.020 m2 to 0.495 ± 0.032 m2. For predictions of AP based on anthropometric algorithms, except for the Heil (2002) method (P = 0.983), all other methods significantly (p < 0.001) over predicted our digitized AP. These Est AP ranged from 0.399 ± 0.025 m2 to 0.550 ± 0.028 m2. No correlation was found between total mass and AP (r = 0.302, p = 0.208).

Stepwise regression was also performed to predict time trial performance time and power. In this multivariate analysis, the best correlates of level time were the mean field-measured power to Ad (r = − 0.92, p < 0.001, y = − 0.53 × time trial power to drag area + 2358.6). The correlation coefficient for this relationship was not significantly different from that for power to k (r = − 0.92, p < 0.001), which indicates that variations in ambient air density between the different time trial days was unlikely to have had much impact on performance time in our group of subjects. If power at V˙O2 peak and LT are normalized to parameters that affect resistance, the best correlate of performance time was power at V˙O2 peak normalized to Ad (r = − 0.92, p < 0.001, y = − 0.48 × power at V˙O2 peak to Ad + 2394.8). This correlation was not significantly different from that obtained using the mean field-measured power normalized to Ad or k. For power output during the time trial, the best correlate from all the variables measured in the laboratory is V˙O2 peak (l min−1) (r = 0.83, p < 0.001, y=54.67×V˙O2 peak + 48.8).

Discussion

As hypothesized, a cyclist’s mean field-measured power output normalized to our field-determined aerodynamic resistance (i.e., k or Ad) was the best correlate of level time trial performance time (r = − 0.92, p < 0.001). These results demonstrate and confirm that aerodynamic resistance is the primary resistance that is faced when cycling on level terrain. Accordingly, lab-measured physiological indices of performance (e.g., power at V˙O2 peak), though significantly correlated to field-measured power output (r = 0.64), were not significantly related to performance time (r = − 0.43), unless normalized to aerodynamic resistance (r = − 0.92). Of note, our field-determined drag area alone was the single best correlation with performance time on the level (r = 0.85, p < 0.001), confirming that aerodynamic resistance in our population played the most significant role in level time trial performance. Further, our field-determined drag area improved correlations to performance time better than either AP or Est AP. In addition, rolling resistance was not a distinguishing predictor of performance reflecting its smaller contribution to the resistance to forward motion during cycling.

In agreement with Balmer, Davison & Bird (2000), we found the majority of laboratory measures were not significantly correlated to level time trial performance time. In contrast to our findings though, a number of investigators have shown stronger correlations between laboratory measures of performance and level time trial performance in the field (Hawley & Noakes, 1992; Nichols, Phares & Buono, 1997; Smith, Dangelmaier & Hill, 1999; Anton et al., 2007). It is likely that these conflicting findings are due to differences in the subject populations recruited. For example, in the study by Anton et al. (2007), the subject population had a relatively homogenous frontal surface area (0.35 ± 0.02 m2) and a relatively heterogeneous maximal mechanical power output (490 ± 56 Watts) compared to the subjects from our study (0.34 ± 0.05 and 362 ± 30, respectively). This homogeneity in frontal surface area likely resulted in a greater correlation between laboratory measures and level time trial performance. Because level time trial performance is essentially a balance between resistive forces (primarily aerodynamic resistance) and propulsive forces (mechanical power output), homogeneity in one factor results in the other heterogeneous factor having a greater correlation. Furthermore, it is our belief that correlations in these previous studies would be stronger if the laboratory performance measures were normalized to the subject’s actual aerodynamic drag area. For instance, in our study, power at LT had a correlation of −0.45 with performance time but when this same laboratory measure was normalized to Ad, the correlation became −0.85.

Our finding that level time trial performance time is best predicted by normalizing field-measured power output to either k or Ad (r = − 0.92, p < 0.001), while perhaps self-evident, underscores the significant absence of actual aerodynamic measures in studies attempting to predict cycling time trial performance in the field. Martin et al. (2006) found measured aerodynamic drag can be used to predict sprinting speed in cyclists. Our study furthers these findings, and is the first to highlight the importance of measured aerodynamics during longer duration events at slower speeds compared to those seen during sprinting. In our study, the mean field-measured power was significantly correlated to level time (r = − 0.59, p < 0.01) but only explained 35% of the variability in level time. However, when field-measured power was normalized to aerodynamics, 85% of the variability was explained by highlighting that differences in aerodynamics are a critical performance determinant.

We found aerodynamic drag alone represented as either k or Ad, was significantly correlated to performance time (r = 0.85, p < 0.001), explaining 72% of the variability in performance time. The heterogeneous equipment and positions subjects used in the time trial may have aided this finding. Out of our nineteen subjects, nine used time trial bicycles, four used standard road bicycles with aerodynamic handlebars, and the remaining six used a standard road bicycle with no specific aerodynamic equipment. However, while each group had significantly different mean values for Ad, within the nine subjects using time trial bicycles, our results were similar with mean field-measured power normalized to aerodynamics the single best predictor of performance time (r = − 0.88, p < 0.001), aerodynamics the next best predictor (r = 0.72, p < 0.001), and field-measured power not predicting performance time (r = − 0.33, p > 0.05). The range in Ad between subjects riding a time trial bicycle (0.258 to 0.355 m2) may have also aided in this finding. This range in Ad though, also highlights the variability that exists between individuals with similar equipment and the importance of determining drag area to predict level time trial performance.

Because aerodynamic drag has been difficult to measure in the past, many studies have attempted to use estimates of the projected frontal area (Est AP) or actual measures of projected frontal area (AP) to represent aerodynamic drag (Ad) (Swain, 1994; Anton et al., 2007). Estimating Ad in this manner assumes a constant coefficient of drag (Cd), which along with AP makes up Ad. However, in the present study, AP only accounted for 54% of the variability in Ad. This discrepancy is largely due to individual differences in Cd. Assuming a constant Cd requires that riders must be riding in the same environmental conditions while using the same bike, geometry, clothing, and equipment. However, the range of Cd values between our subjects (0.823 to 1.262) highlights the individual variability in this measure. Furthermore, in agreement with previous work (Kyle, 1991; De Groot, Sargeant & Geysel, 1995; Debraux et al., 2011), we found no relationship between AP and Cd (r = − 0.17, p = 0.49) demonstrating their independence and individual importance in determining Ad. Accordingly, when normalizing field measured power output during the time trial, we found that the correlation to performance time was significantly lower using AP (r = − 0.75) or Est AP (−0.71) compared to normalizing using our field-determined Ad (r = − 0.92). Thus, actual projected frontal area or an estimate of projected frontal area does not explain all of the variability in Ad and when used to normalize power is not necessarily better in the prediction of level time trial performance than Ad alone (r = − 0.85).

The results from the present study suggest that field-testing with a power meter may be a better alternative to traditional laboratory testing (i.e., V˙O2 peak, LT, economy) for athletes and coaches wanting to predict time trial performance. The field-measured drag area in combination with field-measured power output provided the best prediction of time trial performance compared to any combination of laboratory testing. Furthermore, we demonstrate that determining drag area does not need to be difficult. Our field-measured determination of drag area was the single best predictor of time trial performance and requires only a power meter, speedometer, and flat road making it far more economical than other tools (i.e., a wind tunnel). Thus, athletes can easily quantify and improve performance in events like level time trials by using our field-measured drag area technique to optimize the relationship between aerodynamic resistance and power output.

In conclusion, to predict cycling performance in the field, a cyclist’s ability to produce power must be considered relative to the forces that resist forward motion. Accordingly, level time trial performance is best predicted by normalizing the mean field-measured power output to aerodynamic resistance per velocity squared (k) or the drag area (Ad). Despite the fact that physiological variables are commonly thought of as the most important determinant of performance, we found that alone, measures of aerodynamic resistance were better related to level performance than V˙O2 peak, LT, or economy. Moreover, the use of projected frontal area (AP) or estimates of projected frontal area (Est AP) are not as accurate as directly measured Ad.

Supplemental Information

Appendix S1 Methodological details

Click here for additional data file.

Appendix S2 Raw data from the study

Click here for additional data file.

Additional Information and Declarations

Competing Interests

Author Contributions

Human Ethics

The authors declare there are no competing interests.

James E. Peterman analyzed the data, contributed reagents/materials/analysis tools, wrote the paper, prepared figures and/or tables, reviewed drafts of the paper.

Allen C. Lim and William C. Byrnes conceived and designed the experiments, performed the experiments, analyzed the data, contributed reagents/materials/analysis tools, wrote the paper, prepared figures and/or tables, reviewed drafts of the paper.

Ryan I. Ignatz performed the experiments, analyzed the data, contributed reagents/materials/analysis tools, wrote the paper, reviewed drafts of the paper.

Andrew G. Edwards performed the experiments, analyzed the data, contributed reagents/materials/analysis tools, reviewed drafts of the paper.

The following information was supplied relating to ethical approvals (i.e., approving body and any reference numbers):

All subjects were informed of the risks involved with participation in the study and gave written informed consent before participating. The Human Research Committee at the University of Colorado Boulder approved the protocol used for this study (reference number 0600.20).

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
