# Peer review of "Field-measured drag area is a key correlate of level cycling time trial performance"

_PeerJ, doi:10.7717/peerj.1144_

## Round 0.1 · original submission · Minor Revisions

· Academic Editor

Minor Revisions

Please address the issues raised by the two reviewers. In particular, please consider making the method section more concise. A good way of doing so could be to move some of the details to an appendix.

·

Basic reporting

The reference number for the IRB approval is not mentioned.
If there is external funding, the source of funding is not mentioned.
The manuscript otherwise meets the stated criteria for publication for "basic reporting."
It is unclear if or how the raw data for this study would be made available as stated in the criteria.
The figures are clear and relevant. The text is also clear and unambiguous and no edits to the grammar or writing are noted.

Experimental design

The submission fits into the scope of the Journal as a Medical or Health Science - specifically sports medicine/human performance.

The article also meets the criteria for this section of the review, "Experimental Design." The Research question is clearly defined and supported, and the study conducted with attention to technical accuracy and precision.

The one suggestion for this category is that much of the methods section may be better in an appendix or supplement. The details of the instrumentation could be more concise and, where the calibrations and operations of the instruments do not differ from their recommended use, that information could be moved to an appendix or supplemental table.

Validity of the findings

The results do appear to be robust, statistically sound and well controlled. Again - it is unclear how or if the raw data would be made available.

The conclusions flow logically from the introduction/research question and the results and are clearly explained.

Reviewer 2 ·

Basic reporting

The article is well written, with a sufficient introduction and background to explain the importance of this work.

Perhaps some of the detail in the methods section could be moved to an appendix. This section is quite lengthy. For example, there could be an appendix on calibration of all of the testing equipment. Below are some specific comments.

In Table 2: Did not find the symbol † used anywhere in chart; formatting with subscripts is off in text below table (lines 3-6).

Experimental design

Overall this is clear. Some specifics comments:

Beginning Line 195: It is unclear how Cd is derived in this section.

Line 227: is there a reference for this procedure or is this the first time it has been used as a means to determine AP? You have referenced the other indirect methods.

In several places you state that there was no difference found between observers (line 127-128 and 224-225. Elaborate please.

Validity of the findings

Again, overall this is clear. One specific comment:

Line 227: is there a reference for this procedure or is this the first time it has been used as a means to determine AP? You have referenced the other indirect methods. It appears you are the first to use this direct method.

Additional comments

No comments.

---

## Round 0.2 · accepted · Accept

· Academic Editor

Accept

The authors have responded to all comments made by the reviewers. No additional revision is necessary.